# The Course of Posterior Antebrachial Cutaneous Nerve: Anatomical and Sonographic Study with a Clinical Implication

**DOI:** 10.3390/ijerph18157733

**Published:** 2021-07-21

**Authors:** Jose García-Martínez, Maribel Miguel-Pérez, Albert Pérez-Bellmunt, Sara Ortiz-Miguel, Ginés Viscor

**Affiliations:** 1Horta Osteopathic Clinic, 08031 Barcelona, Spain; 2Unit of Human Anatomy and Embryology, Department of Pathology and Experimental Therapeutics, Faculty of Medicine and Health Sciences (Bellvitge Campus), University of Barcelona, 08907 Hospitalet del Llobregat, Spain; sortiz@uic.es; 3Basic Sciences Department, Universitat Internacional de Catalunya, 08017 Barcelona, Spain; aperez@uic.es; 4ACTIUM Functional Anatomy Group, 08017 Barcelona, Spain; 5Physiology Section, Department of Cell Biology, Physiology and Immunology, Faculty of Biology, University of Barcelona, 08028 Barcelona, Spain; gviscor@ub.edu

**Keywords:** posterior antebrachial cutaneous nerve, ultrasound and anatomical pathway and relations, possible compression

## Abstract

The course of the posterior antebrachial cutaneous nerve (PACN) was studied via ultrasound (US) and dissection. The aim of this study was to reveal the anatomical relationships of PACN with the surrounding structures along its pathway to identify possible critical points of compression. Nineteen cryopreserved cadaver body donor upper extremities were explored via US and further dissected. During US exploration, two reference points, in relation with the compression of the nerve, were marked using dye injection: (1) the point where the RN pierces the lateral intermuscular septum (LIMS) and (2) the point where the PACN pierces the deep fascia. Anatomical measurements referred to the lateral epicondyle (LE) were taken at these two points. Dissection confirmed the correct site of US-guided dye injection at 100% of points where the RN crossed the LIMS (10.5 cm from the LE) and was correctly injected at 74% of points where the PACN pierce the deep fascia (7.4 cm from the LE). There were variations in the course of the PACN, but it always divided from the RN as an only branch. Either ran close and parallel to the LIMS until the RN crossed the LIMS (84%) or clearly separated from the RN, 1 cm before it crossed the LIMS (16%). In 21% of cases, the PACN crossed the LIMS with the RN, while in the rest of the cases it always followed in the posterior compartment. A close relationship between PACN and LIMS, as well as triceps brachii muscle and deep fascia was observed. The US and anatomical study showed that the course of PACN maintains a close relationship with the LIMS and other connective tissues (such as the fascia and subcutaneous tissue) to be present in its pathology and treatment.

## 1. Introduction

Lateral epicondylitis is a common painful condition, affecting between 4 and 7 per 1000 people annually [1]. It is characterized by pain on the lateral side of the elbow, which increases during gripping or squeezing. The lateral elbow pain is common in athletes and manual workers [2,3]. The identification of the cause of elbow pain is sometimes difficult. Possible causes range from extensor tendinopathy to neuropathy affecting the lateral antebrachial cutaneous nerve or the posterior antebrachial cutaneous nerve (PACN) [4,5]. Cutaneous nerve entrapment syndromes have been studied for a long time [6]. Although it can occur due to external causes, compression via musculotendinous, ligamentous or fascial structures are possible [6,7,8,9]. The cutaneous nerve entrapment plays an important role in neuropathic pain syndrome and the diagnostic testing can include electrodiagnostic studies [6,10], anesthetic injection, magnetic resonance imaging, and US [10].

Numerous sites of peripheral nerve compression have been described along its course. Most are associated with surrounding connective and fascial tissues of the upper arm, as can be the medial antebrachial cutaneous nerve, which develops pain due to repeated minor trauma (full extension of the elbow and repetitive forceful contracture of the flexor musculature) [11], the lateral antebrachial cutaneous nerve where crossing the antebrachial superficial fascia [12] and the superficial branch of the radial nerve (also named Wartenberg’s syndrome). The causes of this nerve entrapment include compression from the brachioradialis and the extensor carpi radialis longus and an anomalous fascial ring [13]. These nerves may be the reason for the forearm pain, but little attention has been paid to the possible role of PACN compression. The PACN, also known as the posterior cutaneous nerve of the forearm, arises from the radial nerve (RN) in the posterolateral arm, proximal to the radial canal [14] and the lateral intermuscular septum (LIMS) [15]. Then, it courses posterolaterally to the LIMS and deep to the lateral head of the triceps brachii (LHTB) muscle. At the junction of the brachioradialis and the extensor carpi radialis longus muscle, the PACN pierces the deep fascia entering into the subcutaneous tissue and further divides into anterior and posterior branches, supplying the skin of the lateral epicondyle [15] and the posterior lateral part of the forearm [6,16]. This nerve and its branches can be observed via ultrasound (US) [6,17] when the transducer is placed in the short-axis view at the posterior mid-arm level. The RN is then seen underneath the LHTB muscle and, by moving the transducer more distally, the PACN is seen to divide from the RN [6]. It subsequently pierces the lateral intermuscular septum of the arm to run in the subcutaneous tissue close to the cephalic vein [17].

To date, anatomic and ultrasound studies regarding the topography of the PACN have mostly focused either on the surgical approach to avoid its injury or on ultrasound-guided nerve blockade [4,5,15,18,19,20] of the lateral humeral epicondyle [15]. However, studies involving the relations that this nerve could have with other anatomical structures (such as the superficial and deep fascia) are lacking. This fascia–nerve interplay could have a critical relation with some elbow pains, and some authors have described the target fascia as an important component of treatment in the field of manual treatment and rehabilitation [21]. The aim of this study was to examine the course of the PACN via ultrasound and anatomical dissection and to describe all its topographical relationships (specifically with the LIMS and fascial tissue), thus allowing for a better understanding of possible points of compression and aiding in a more accurate approach to interventional therapy.

## 2. Materials and Methods

This study included 19 cryopreserved shoulder-upper limbs (9 right and 10 left) from 15 cadavers (9 from women and 6 from men, ages 68–91 years old) without evident pathological findings (such as injury or previous surgery). Each specimen was accurately labeled with reference to the side, age and sex of the donor. All the procedures described in this report were approved by the Comité de Bioética of the Universitat de Barcelona (Institutional Review Board IRB0003099) and followed the relevant guidelines and local regulations. The Body Donation Service to Science and dissection room of the Faculty of Medicine and Health Sciences (Campus Bellvitge) provided the cryopreserved upper limbs of the University of Barcelona. They corresponded to donors who, at the time of the donation, did not explicitly oppose the use of their bodies for the study of the anatomy of the upper limbs and who had voluntarily donated their bodies for teaching and research purposes.

### 2.1. Ultrasonography Study

All examinations were performed by the same expert sonographer specialized, who has more than 10 years of experience in musculoskeletal US using a high-frequency linear transducer (5–12 MHz) connected to a US system (Logiq P5, General Electric). In order to prove the accuracy of the ultrasound imaging and to correlate it with the findings in the posterior anatomical dissections, a series of US-guided infiltrations were carried out. These infiltrations were performed using a 1-mL syringe and a 25-G needle to inject 0.01 to 0.05 mL of green or red ink (colorant CLQ -540) adjacent to the nerves. The targeted points were located where the RN pierces the LIMS and where the PACN pierces the deep fascia to become superficial. The distances from the points of injection to the lateral epicondyle, as a landmark were superficially measured using a digital caliper (Figure 1 and Table 1).

The mean value of three consecutive measurements was calculated for each site.

The following US technique was used to locate the regions of interest; first a posterior approach to the arm with the elbow in pronation and 90º flexion was taken. The probe was oriented in the short-axis view. Then, the RN was located in the radial canal, followed by proximal-to-distal scanning to look for the PACN with the probe. Finally, the division of the PACN from the RN was observed.

### 2.2. Anatomical Study

Following the US study, we dissected 19 upper limbs in the same position as US. Dissection proceeded in a proximal distal direction, from the shoulder to the forearm (ending at the wrist). A longitudinal superficial incision was performed through the midline of the upper limb along the lateral side of the arm (centered over the LIMS) and extended distally and posteriorly toward the elbow and forearm, respectively, followed by a horizontal cut in the elbow area to allow for complete dissection. The skin and subcutaneous adipose tissue of the superficial fascia were removed in layers, exposing the brachial/antebrachial deep fascia. Between the superficial fascia and deep fascia, the PACN was identified and exposed in its entirety (including its distal branches), and the surrounding connective and fascial tissues surrounding were studied. The dissection localized the previous injected dye relative to the anatomically identified target point of RN and PACN.

After isolation, measurements of RN and PACN were taken at four different points defined by anatomic landmarks or clinically important points as shown in Figure 1. All of the distances were taken from the lateral epicondyle as landmark: Arrow A; to the point where the RN pierces the LIMS. Arrow B; to the point where the PACN pierces the deep fascia to become superficial. Arrow D; the bifurcation of the PACN in the anterior (A-PACN) and posterior (P-PACN) branches. Finally, the anterior/perpendicular distance from the A-PACN and P-PACN to the lateral epicondyle. The diameter (Ø) of PACN when piercing the deep fascia in the arm were measured (Point C) (Table 1). The measurements were measured using a digital caliper. The mean values of three consecutive measurements were calculated for each parameter. All values are expressed as mean ± and standard deviation. Student’s t-test was used to detect any significant differences in the measurement data between different genders and sides. We analyzed the agreement between US and anatomical measurements by applying the Bland–Altman graphical method (difference plots).

## 3. Results

The ultrasound study showed the ability to correctly identify the PACN, as further confirmed by dissections. However, the PACN presented a certain degree of individual variability in its anatomical course.

### 3.1. Ultrasonography Study

The ultrasound image of the RN as an ovoid shape with a typical fascicular echo texture was identified in the radial canal. It was easily identified in the short-axis view, when it crossed this canal in contact with the humerus bone (Figure 2).

It was recognized correctly in all cases as confirmed by dissection. This was the best landmark to use to find the PACN, which was identified as a division from the RN. Following division, the RN pierced the LIMS 10.5 cm (Range = 3.9 cm, maxim value 12.3 cm and minim value 8.4 cm) from the LE and passed on into the anterior compartment of the arm. The PACN, once divided from the RN, crossed the deep fascia 7.4 cm (Range *=* 6.6 cm, maxim value 11.6 cm and minim value 5 cm) from the LE. US images showed that the nerve was inside a tunnel formed by connective and adipose tissues (Figure 3).

This tunnel was a clear sonographic landmark for locating the nerve and following it until it reached the posterior side of the forearm.

### 3.2. Anatomical Study

Dissection showed the RN in the radial canal, before and after it crossed the LIMS. Ultrasound injected dye confirmed this in 100% of cases.

We initially identified the PACN superficially in the subcutaneous tissue and after examined its deep course. Figures 5 and 6 shows the schematized lines of the different pathways of the PACN, always divided as a branch from the RN, but then it either ran close and parallel to the RN until the RN crossed the LIMS (16 cases, 84%; Lines A to C) or, alternatively, it ran clearly separated from the RN, 1 cm before it crossed the LIMS (3 cases, 16%, Line D) (Figure 4A and Figure 5).

In the first variation (16 cases), the PACN followed the posterior compartment in twelve cases (63%; Line C) (Figure 4B and Figure 5), but in the remaining four cases (21%), the PACN crossed the LIMS with the RN. In two of these latter cases, PACN returned to the posterior compartment (Line B), following its more frequent course, but in the other two of these cases, it followed the RN into the anterior compartment until it crossed the deep fascia on the anterior side of the forearm and finished as a single branch in the forearm (Line A) (Figure 5) or anastomosed with the posterior brachial cutaneous nerve (Figure 6).

An accurate description of the posterior compartment of the arm revealed that in the most frequent cases, when the RN and the PACN ran parallel to each other, both nerves were always located lateral to the LHTB muscle until the RN crossed the LIMS 11.3 cm (±1.1) from the lateral epicondyle. In this arrangement, longitudinal aponeurotic bands from the LHTB to the LIMS formed a longitudinal tunnel shared by both nerves (Figure 7).

An important observation was that on the superficial side of this tunnel, the origin of the muscle fibers of the LHTB muscle formed the roof of the tunnel. Once the PACN had separated from the RN in 12 cases (63%), it ran along the posterior side of the LIMS for 3 cm until it crossed the deep fascia 8.3 cm from the lateral epicondyle (Figure 4B). At this point, it was identified in the adipose tissue of the superficial fascia and had a diameter of 1.2 mm (Table 1).

Lax connective tissue from this superficial fascia formed a structure similar to a canal until it reached the proximal part of the elbow. This anatomical view was compatible with the ultrasound image described earlier (Figure 3). Dissection of the PACN was easiest from the elbow until the PACN terminated, because it had a close relation with the antebrachial fascia.

In the three cases in which PACN clearly separated from the RN (16%) both nerves ran inside the muscular compartment of the triceps brachii muscle, in a canal formed by its perimysium until it crossed the muscle (Figure 4A).

Dye indicating the point of exit of the PACN from the deep fascia into the superficial fascia was correctly injected in 74% of cases and 7.4 cm from the lateral epicondyle. However, in 10% of cases, dye was found in target points along the course of the PACN, and in the remaining 16% of the specimens, it was injected into the posterior brachial cutaneous nerve. The latter run anteriorly with the RN (Figure 5).

During the PACN’s descent, and from 5.7 cm superior to the lateral epicondyle, it usually divided into two branches, forming the anterior PACN (A-PACN) and the posterior PACN (P-PACN), but in one of the specimens the PACN divided into three branches. In all cases (100%), after division and before the nerve reached the elbow, the PACN branches became anterior, passing anterior to the lateral epicondyle (Figure 8).

The A-PACN descended as a single branch in 95% of cases (*n* = 18). Of those, 89% (*n* = 16) finished on the posterior side of the wrist and 11% (*n* = 2) only reached the distal third of the forearm. This branch had a perpendicular distance of 3.6 cm from the lateral epicondyle. In the other 5% (*n* = 1), the A-PACN divided into two branches: one finished halfway along the forearm’s posterior side and the other reached the wrist.

The P-PACN as a single branch (87%) could anastomose with the posterior brachial cutaneous nerve and then, in few cases (16%), descend until the middle third of the forearm, or finish in the skin of the lateral epicondyle (42%) or finish in the inferior part of the lateral epicondyle (42%). When P-PACN divided into two branches (13%), one finished in the lateral epicondyle and the other reached the posterior side of the forearm (Figure 8).

Dissection of the LIMS showed that it was a dense connective tissue that was anchored to the lateral border of the humerus and separated the anterior and posterior compartment of the arm; macroscopic observation of the connective tissue showed an arrangement in different directions and a close relation with the origin of the arm muscles. However, there was a large amount of variability regarding the origin of the posterior (LHTB) and anterior muscles of the arm (brachialis, brachioradialis, extensor carpi radialis longus). This variability could be divided into two types: Type 1: insertions of muscles of less than 3 cm long (33%); or Type 2: insertions of muscles over 3 cm long (67%). The LIMS had a close relation with the RN and PACN via the tunnel described earlier between the LIMS and the triceps brachii muscle.

## 4. Discussion

This study shows that the PACN can be correctly identified via an ultrasound exploration, as confirmed by posterior dissection. In addition, the measurements obtained by US are in good agreement with further anatomical dissection measurements as can be seen in the Bland–Altmann graphical analysis (Figure 9).

Some previous studies accurately described the course of the PACN using US [6,17] and dissection [17,22], but few details are known about its relation with other anatomical structures such as the LIMS, and the superficial and deep fascia or muscles. Both of these aforementioned studies focused on the PACN in the context of forearm-based neuropathic pain syndromes [17] or in US-guided diagnostic and interventional procedures targeting the nerve in selected patients [22]. The current study revealed the PACN’s close relation with the LIMS and the deep and superficial fascia. We investigated this relation because they could be relevant in pathogenesis via compression of the PACN during its course and it could be relevant for its treatment with manual therapies or nerve release. In this context, some authors have already reported the use of the mobilization-with-movement treatment technique as a combination of manual therapy (manual sustained lateral glide of the elbow) and therapeutic exercise (repeated gripping action) for the treatment of some cases of lateral epicondylitis in order to avoid surgery [23].

Sonoanatomically, our finding that the RN passed through the LIMS 10.5 cm from the lateral epicondyle was similar to other reports of a 10 cm distance [22]. Likewise, the PACN always originated from the RN on the posterior side of the arm and before the RN crossed the LIMS [22,24,25]. The frequency of the posterior course of the PACN (78%) was similar to that reported by Hannouche D et al., (83%) [5]. As described herein, and consistent with other authors, the PACN ran parallel to the RN and maintained a close relation with the LIMS [22,25]. We also describe herein a novel tunnel formed by aponeurosis fibers from the triceps brachii muscle to the LIMS. The results suggest that the fibrous connective tissue that forms this tunnel could be involved in the possible compression of the nerves. When the PACN did not run parallel to the RN (16%) (Line D/Figure 5) it followed a divergent course from the RN, crossing the muscular compartment of the triceps brachii muscle inside the perimysium. In this case, the PACN could be subject to compression by the triceps brachii muscle, thus causing neuropathic pain in the arm and forearm.

In these two possible compression scenarios, the application of manual techniques would be aimed at releasing possible muscular–fascial tensions and, therefore, would be useful for the treatment of compression. Knowing the relation of the PACN with the LIMS–LHTB would allow us to understand the risk of entrapment and irritation of the PACN and that is the reason why these techniques must be carried out as a matter of priority in this septum. Manual treatment would then be aimed mainly at reducing the muscular tensions in the LIMS and the related muscles (LHTB, brachial and brachioradialis). However, the focus should be directed to the LHTB, since our results showed that it had the highest amount of muscle insertion in the septum, thus having the greatest influence on the tensions exerted on it. The objective would be to balance the forces that reach the LIMS (anterior and posterior) by improving its tension and avoiding its densification and, therefore, the loss of viscoelasticity [26] that usually leads to problems arising along the course of vascular–nervous structures [27]. It has been shown that the aponeurotic orifice of the nerve is usually indurated when there is a fixation [28], the fixation in this study being the insertion of the above mentioned muscles. Other studies have confirmed the effectiveness of myofascial treatment in epicondylalgia [29].

The PACN is considered to be mainly a sensory nerve, but also drives a considerable amount of efferent sympathetic signals [30]. This supports the hypothesis that these subcutaneous nerves may play a role in the pathogenesis of complex regional pain syndrome, and there have been reports about induction of complex regional pain after injury to these nerve [30].

The point of exit of the PACN from the posterior compartment of the arm through the deep fascia was 7.4 cm from the lateral epicondyle as measured ultrasonographically and 8.3 cm anatomically. Some studies have reported similar distances of between 7.24 cm [18] and 7 cm [22] but others have reported shorter distances such as 6.6 cm [10] and even 5 cm [20]. Possible explanations for these differences could be the landmarks used for the measurements, individual variations and the difficulty in differentiating between the superficial and deep fascia. The transition zone of the nerve, from the deep fascia to the superficial fascia (forming a fascial ring), is an area that could be affected by fascial tension changes or by an imbalance in muscular tension caused by lateral epicondylitis of long evolution or repetitive mechanical work of the upper limb. Such circumstances cause an increase in the size and density of the connective tissue of both the deep and superficial fascia [31], which may hinder the sliding of the PACN with consequent irritation. In cases of fixation, the nerve elongates with difficulty during arm movement, causing sensitivity or pain [27,28,30]. Our description is similar to the proposal of Maida et al. [22], who described these nerves as monofascicular hyperechoic dots within the subcutaneous tissues.

The difference between the measurements obtained using the ultrasound and anatomy methods (bias) can be explained by small morphological modifications arising during the dissection of the upper limb. As a result of the mechanical actions taken to separate the tissues, the distances between the reference points could increase, or these differences could be due to limitations in the accuracy of US as well.

After the nerve reached the superficial fascia, the PACN followed an anterior pathway to the lateral epicondyle without exception. However, this finding contrasted with those other authors who described localization posterior to the lateral epicondyle [22,25].

The relation with other structures (such as the cephalic vein) was parallel and volar except in the proximal part of the forearm, which is inconsistent with other reports [17,18]. We found that the PACN was deeper to the vein in the superficial fascia in all cases until reached the distal side of the forearm (Figure 9). This is important for surgery in this area.

A previous study posited that the PACN divides into two branches after it crosses the deep fascia 7 cm from to LE [22]. We observe the two branches anterior to the LE at the same level, whereas the abovementioned study reported that the posterior division travelled further before crossing the deep fascia and also had a major bifurcation with a posterior branch extending to the anconeous muscle [22]. However, another study found few posterior divisions of the PACN [25].

This study has some limitations, such as those due to the intrinsic technical constraints of US exploration. This investigation was a description of possible fascial structures that can give a compression of the PACN. Futures clinic and ultrasound researches could describe the implication of the fascial structures in the nerve compression and help in its diagnosis and treatment.

## 5. Conclusions

A detailed knowledge of the PACN pathway and its anatomic relationships with the surrounding structures could help to determine the cause of pain in this area. This study has showed fascial and musculoskeletal structures that could play an important role on this kind of compression.

## Figures and Tables

**Figure 1 ijerph-18-07733-f001:**
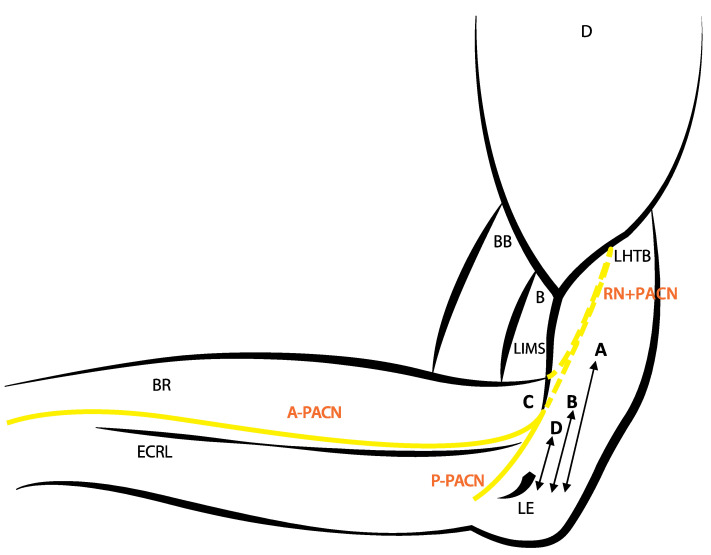
**D** (deltoideus muscle), **BB** (biceps brachii muscle), **BR** (brachioradialis muscle), **B** (brachialis muscle), **LHTB** (lateral head of the triceps brachii), **ECRL** (extensor carpi radialis longus muscle), **LIMS** (lateral intermuscular septum), **RN** (radial nerve), **PACN** (posterior antebrachial cutaneous nerve), **A-PACN** (anterior branch of the PACN), **P-PACN** (posterior branch of the PACN) and **LE** (lateral epicondyle). **Arrow A**: Distance from the LE to the point where the RN pierces the LIMS. **Arrow B**: Distance from the LE to the hiatus where the PACN pierces the deep fascia. **Point C:** PACN diameter in the point where the nerve pierces the deep fascia in the arm. **Arrow D**: Distance from the LE to the bifurcation of the PACN in the A-PACN and P-PACN.

**Figure 2 ijerph-18-07733-f002:**
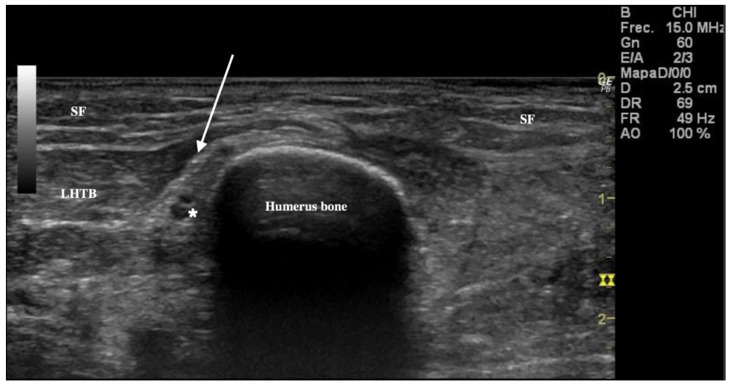
Ultrasound of the aponeurotic part of lateral head triceps brachii (LHTB) (white arrow) that forms a fibrotic tunnel for the radial nerve (RN) (*) and posterior antebrachial cutaneous nerve before the RN crosses the lateral intermuscular septum, followed by the superficial fascia or subcutaneous fatty layer (SF).

**Figure 3 ijerph-18-07733-f003:**
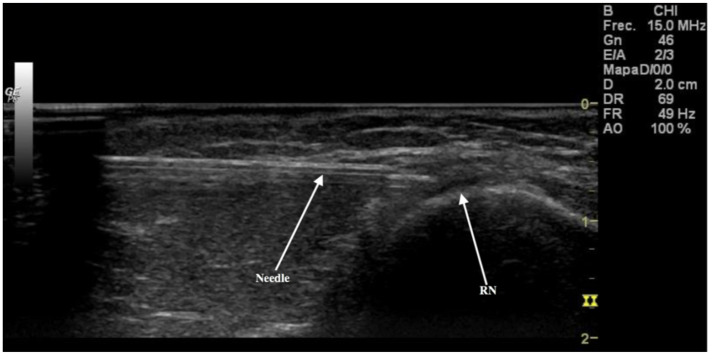
Ultrasound image of the radial nerve (RN) (white arrow) that is in contact with the humerus bone. Needle injected dye in the nerve point before crossing the lateral intermuscular septum.

**Figure 4 ijerph-18-07733-f004:**
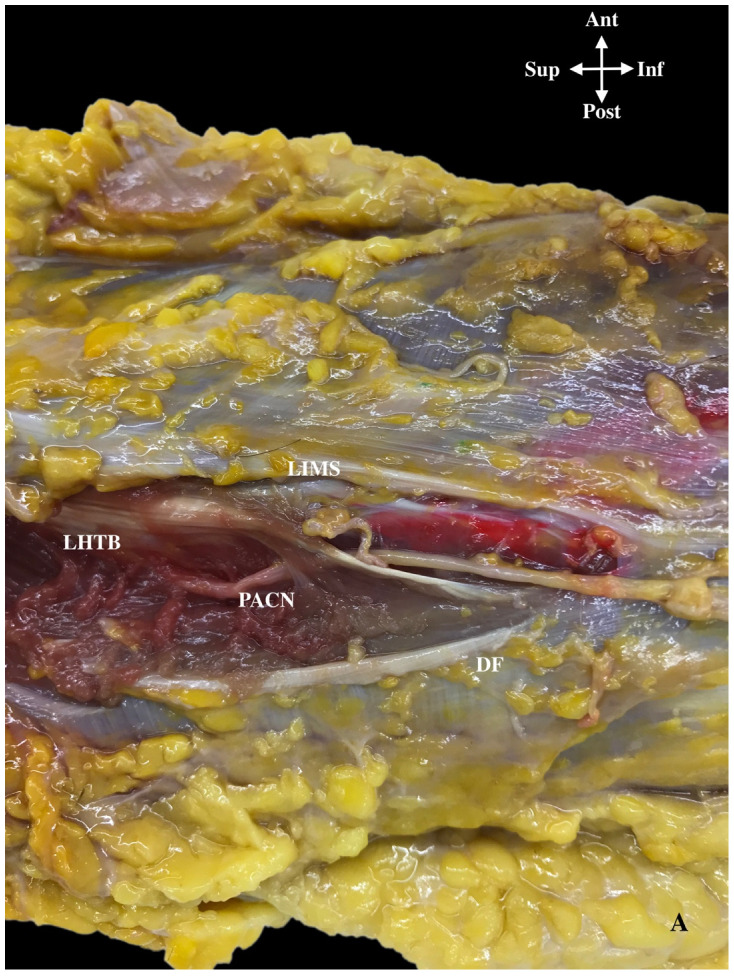
(**A**) Posterior antebrachial cutaneous nerve (PACN) separated from the radial nerve (16%) that in the picture is deep to the LHTB longitudinal aponeurosis, just before it crosses the LIMS (16%). PACN ran alone inside the muscular fibers of the triceps brachii muscle in a canal formed by the perimysium and aponeurotic bands of this muscle. Lateral intermuscular septum (LIMS), lateral head triceps brachii muscle (LHTB), deep fascia (DF). (**B**) Common and classical course of the PACN (white arrows) that ran along the posterior side of the lateral intermuscular septum (LIMS) until it pierced the deep fascia in this posterior compartment (63%) and divided into two branches, an anterior PACN (A-PACN) and a posterior PACN (P-PACN). Brachialis muscle (B), brachioradialis muscle (BR), lateral epicondyle (LE).

**Figure 5 ijerph-18-07733-f005:**
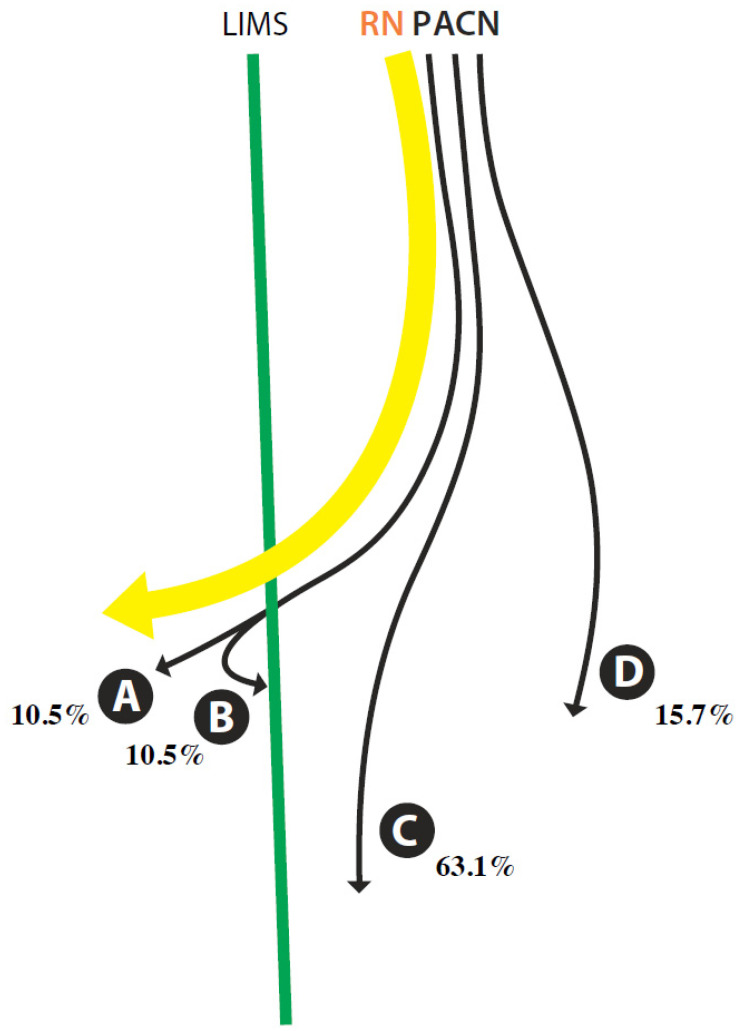
Drawing of the different pathways of the posterior antebrachial cutaneous nerve (PACN) in the arm where its divided from radial nerve (RN): Line A: PACN piercing the lateral intermuscular septum (LIMS) with the radial nerve (RN) and it follows in the anterior compartment until reaches the skin (10.5%). Line B: PACN pierces the LIMS with the RN and come back to the posterior compartment until reaches the skin (10.5%). Line C: More frequently pathway of the PACN in the compartment posterior (63.1%). Line D: clearly separated from the RN, 1 cm before it crossed the LIMS (15.7%).

**Figure 6 ijerph-18-07733-f006:**
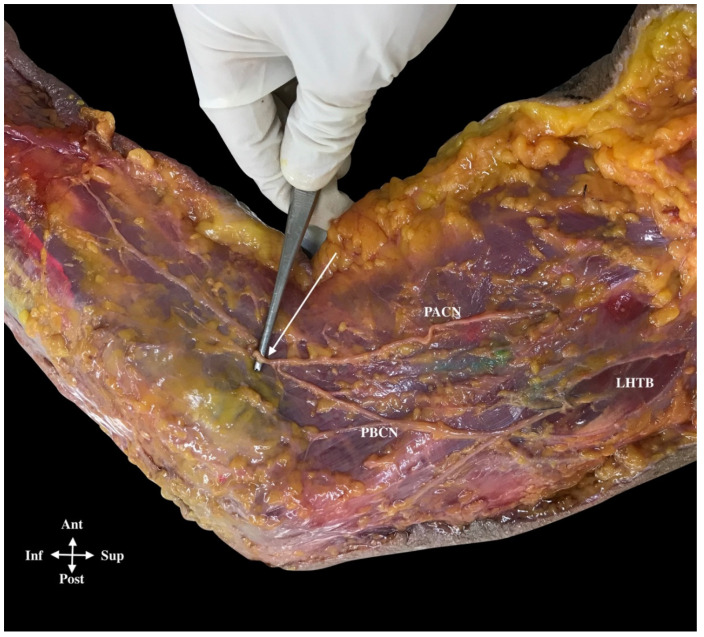
Anastomosis between the posterior antebrachial cutaneous nerve (PACN) (piercing the LIMS with the RN and then following in the anterior compartment) and the posterior brachial cutaneous nerve (PBCN). They form the only branch in the posterolateral side of the forearm (white arrow). Lateral head triceps brachii muscle (LHTB).

**Figure 7 ijerph-18-07733-f007:**
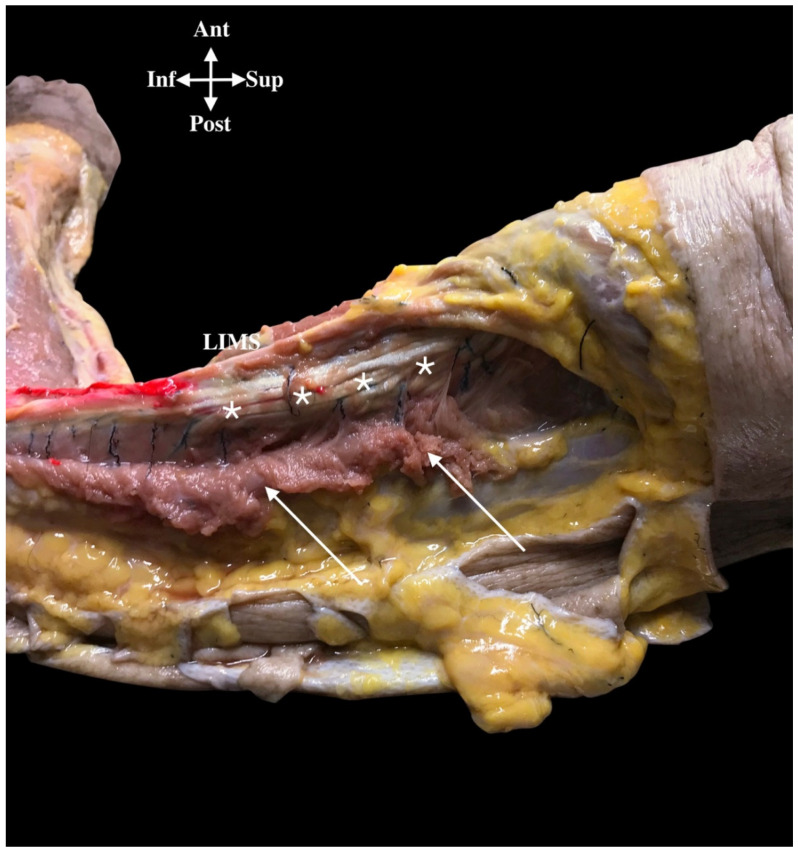
The picture shows the longitudinal aponeurotic fibers (*) from the lateral head triceps brachii muscle (LHTB) after dissecting the muscular fibers of the LHTB (white arrows). The posterior antebrachial cutaneous nerve pierces the deep fascia in the posterior compartment (red dye). Lateral intermuscular septum (LIMS).

**Figure 8 ijerph-18-07733-f008:**
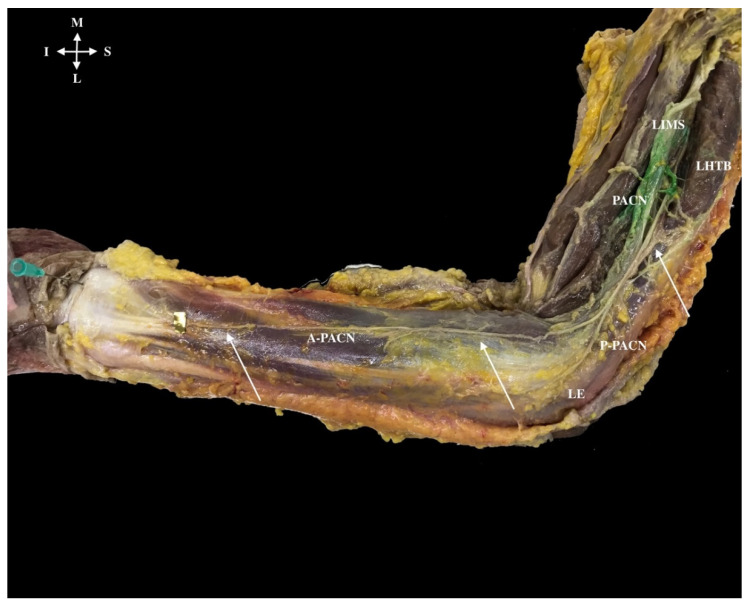
Pathway of the posterior antebrachial cutaneous nerve (PACN) (superior white arrow) after crossing the deep fascia of the arm (green dye). Posterior division into two branches anterior (A-PACN) (inferior white arrows) and posterior (P-PACN) was always anterior to the lateral epicondyle (LE). The needle landmarks the wrist. Lateral intermuscular septum (LIMS), lateral head triceps brachii muscle (LHTB).

**Figure 9 ijerph-18-07733-f009:**
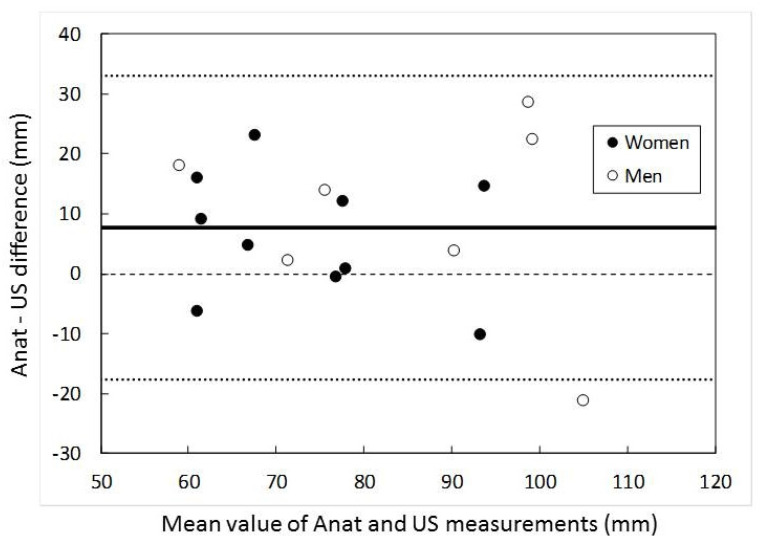
Bland–Altmann graphs for the distances from lateral epicondyle to the hiatus point where the posterior antebrachial cutaneous nerve (PACN) pierces the deep fascia. The Y-axis represents the difference in the measurements taken by anatomical dissection and US imaging. The X-axis represents the mean value of the measurements obtained by the two methods. The horizontal solid line represents the bias, whereas the dotted lines are the upper and lower limits of agreement (LOA), calculated as 1.96 SD (standard deviation). A dashed line indicates zero difference (equality).

**Table 1 ijerph-18-07733-t001:** Ultrasonographyc (US)/Anatomic (A) diameter/distance of the different landmarks between the RN and PACN to LE.

	A (US)	A (A)	B (US)	B(A)	C	D	E	F
Mean (mm)	105	113	74	83	1.2	57	36	27
SD (mm)	8	11	13	13	2	12	8	6
Range (mm)	123–84	138–99	116–50	113–58	16–7	86–37	58–23	33–17

Values are presented as mean (SD, standard deviation); Abbreviations: LE, lateral epicondyle; RN, radial nerve; LIMS, lateral intermuscular septum; PACN, posterior antebrachial cutaneous nerve; A-PACN, anterior; P-PACN, posterior; Distances: A: From the LE to the point where the RN pierces the LIMS. *B:* From the LE to the hiatus where the PACN pierces the deep fascia. *C:* PACN diameter at the point where the nerve pierces the deep fascia in the arm. *D*: From the LE to bifurcation of PACN in A-PACN y P-PACN. *E:* The distance anteriorly/perpendicular where the A-PACN passes the LE. *F:* The distance anteriorly/perpendicular where the P-PACN passes the LE.

## Data Availability

The data that support the findings of this study are available from the corresponding author, M.M.P., upon reasonable request.

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
