# Peer review of "The Course of Posterior Antebrachial Cutaneous Nerve: Anatomical and Sonographic Study with a Clinical Implication"

_ijerph, 2021, doi:10.3390/ijerph18157733_

Round 1

Reviewer 1 Report

In this study, the authors studied the course of the posterior antebrachial cutaneous nerve via ultrasound and anatomical observation in 19 upper limbs. The research objectives are well defined. The study methodology in general is appropriate and clear. The results are well described. The figures are high-quality and informative.

The discussion section is adequate, and the clinical implications of the study are well reported. The written English in general is fine, although I believe some sentences need to be clarified (see comments below). There are also some details that need to be explained.

I believe this study presents results useful for clinicians and physiotherapists, and in general has been well conducted. I hope my comments below could help improve the quality of this paper.

  1. In table 1, the ranges include only the minimal values. Additionally, the caption reports that the standard deviations are included, but I cannot find any standard deviation in the table or the text.
  2. Several times the authors report actual measures in the text. For example “both nerves were always located lateral to the LHTB muscle until the RN crossed the LIMS 11.3 cm from the lateral epicondyle.” It would be helpful to include in the actual text the standard deviations of these average measures.
  3. “Student’s t test was used to detect any significant differences in the measurement data between different genders and sides.” This analysis is not reported in the manuscript.
  4. Please specify what type of dye was used.
  5. “We found that the PACN was deeper to the vein in the superficial fascia in all cases until 375 reached the distal side of the forearm.“ It would be great to include one or a few figures showing the relation between nerves and blood vessels.
  6. “Cutaneous nerve entrapment syndromes have long been overlooked as a cause [4], although can occur due to external compressions, [4,31] around musculotendinous, ligamentous structures, narrow osteofibrous tunnel, an opening in a fibrous bands, or an aponeurotic ring, because of the potential for increased strain and compression on the peripheral nerve at those sites”. This sentence is not clear. Please rephrase.
  7. “the RN pierced the LIMS 10.5 cm (R= 3.9 cm) from the LE… crossed the deep fascia 7.4 cm (R= 6.6 cm)”. It is not clear what the 2 Rs mean.
  8. “In two of these latter cases, PACN returned to the posterior compartment (Line B), following its more frequent course, but in the other two of these cases, it followed the RN in the anterior 217 compartment until it crossed the deep fascia on the anterior side of the forearm and finished as a single branch in the forearm (Line A) or anastomosed with the posterior brachial 219 cutaneous nerve (Fig. 5).” It’s not clear what the Lines A and B are.
  9. “However, it has been reported that the application of manual techniques in other similar conditions can be useful.” It is not clear what this statement means in this context.
  10. “Fortunately, the latter point can be easily identified echographically within the channel formed by the connective tissue of the superficial fascia.” Please clarify what the latter point is.

Author Response

In this study, the authors studied the course of the posterior antebrachial cutaneous nerve via ultrasound and anatomical observation in 19 upper limbs. The research objectives are well defined. The study methodology in general is appropriate and clear. The results are well described. The figures are high-quality and informative.

The discussion section is adequate, and the clinical implications of the study are well reported. The written English in general is fine, although I believe some sentences need to be clarified (see comments below). There are also some details that need to be explained.

I believe this study presents results useful for clinicians and physiotherapists, and in general has been well conducted. I hope my comments below could help improve the quality of this paper.

In table 1, the ranges include only the minimal values. Additionally, the caption reports that the standard deviations are included, but I cannot find any standard deviation in the table or the text.

Thank you very much. We change the table 1 and we have added this information.

Table 1. Ultrasonographyc (US)/Anatomic (A) diameter/distance of the different landmarks between the RN and PACN to LE

A (US)

A (A)

B (US)

B(A)

C

D

E

F

Mean (mm)

105

113

74

83

1.2

57

36

27

SD (mm)                 

8

11

13

13

2

12

8

6

Range (mm)                  

123–84

138–99

116–50

113–58

16–7

86–37

58–23

33–17

Values are presented as mean (SD, standard deviation)

Abbreviations: LE, lateral epicondyle; RN, radial nerve; LIMS, lateral intermuscular septum; PACN, posterior antebrachial cutaneous nerve; A-PACN, anterior; P-PACN, posterior

Distances: A: From the LE to the point where the RN pierces the LIMS B: From the LE to the hiatus where the PACN pierces the deep fascia C: PACN diameter in the point where the nerve pierces the deep fascia in the arm D: From the LE to bifurcation of PACN in A-PACN y P-PACN E: The distance anteriorly/perpendicular where the A-PACN passes the LE F: The distance anteriorly/perpendicular where the P-PACN passes the LE.

Several times the authors report actual measures in the text. For example “both nerves were always located lateral to the LHTB muscle until the RN crossed the LIMS 11.3 cm from the lateral epicondyle.” It would be helpful to include in the actual text the standard deviations of these average measures.

Thank you very much. We have added this information in the text. The SD is 1.1 cm. Also this information is in the table.

 “Student’s t test was used to detect any significant differences in the measurement data between different genders and sides.” This analysis is not reported in the manuscript.

Thank you very much for your observation, we are very sorry, we have added this information in the text.

Please specify what type of dye was used.

We are sorry, we have already specified.

“We found that the PACN was deeper to the vein in the superficial fascia in all cases until 375 reached the distal side of the forearm.“ It would be great to include one or a few figures showing the relation between nerves and blood vessels.

Thank you very much for your suggestion.  It is would be a good idea however we would have to made more changes in all the article and we only wanted to remark that the nerve is also in the subcutaneous tissue but it is deeper to the superficial veins.

“Cutaneous nerve entrapment syndromes have long been overlooked as a cause [4], although can occur due to external compressions, [4,31] around musculotendinous, ligamentous structures, narrow osteofibrous tunnel, an opening in a fibrous bands, or an aponeurotic ring, because of the potential for increased strain and compression on the peripheral nerve at those sites”. This sentence is not clear. Please rephrase.

Thank you very much for your suggestions. We have rewritten the sentence.

“the RN pierced the LIMS 10.5 cm (R= 3.9 cm) from the LE… crossed the deep fascia 7.4 cm (R= 6.6 cm)”. It is not clear what the 2 Rs mean.

This measures were taken by ultrasound and the meaning of the R is the range but we have replaced the R for the maximum value 10,5 cm (12.3 cm) and the minimum value (8.4 cm) In the case of the deep fascia 7.4 cm, it would be the maximum value (11.6) and the minimum value (5 cm). We add this information in the text.

 “In two of these latter cases, PACN returned to the posterior compartment (Line B), following its more frequent course, but in the other two of these cases, it followed the RN in the anterior 217 compartment until it crossed the deep fascia on the anterior side of the forearm and finished as a single branch in the forearm (Line A) or anastomosed with the posterior brachial 219 cutaneous nerve (Fig. 5).” It’s not clear what the Lines A and B are.

Thank you very much for your observations.  We have added a graphic that resumes the variations of the course of the nerve. It implies that we have had to change the numeration of all the figures.

“However, it has been reported that the application of manual techniques in other similar conditions can be useful.” It is not clear what this statement means in this context.

Thank you very much. We have reformulated this paragraph; we hope that now it is clearer.

“Fortunately, the latter point can be easily identified echographically within the channel formed by the connective tissue of the superficial fascia.” Please clarify what the latter point is.

Thank you very much. This sentence does not provide any important information so we have eliminated it.

Reviewer 2 Report

In this manuscript, the authors performed ultrasonography (US) and cadaveric studies, investigating the anatomical relationships of posterior antebrachial cutaneous nerve (PACN) with surrounding structures to identify possible critical points of compression. They found that PACN course maintains a close relationship with those of the radial nerve (RN), the lateral intermuscular septum (LIMS) and other connective tissues such as the deep and superficial fascia, and that the measurements obtained by US are in good agreement with further anatomical dissection measurement. I have several concerns for this manuscript.

Title:

“Can the course of the posterior antebrachial cutaneous nerve explain some elbow pains?” Your results did not answer this question.

Abstract:

Authors concluded the tight relationships could explain the existence of PACN entrapment points and the possibility of successful elbow pain treatment by applying manual therapy techniques. However, golden standard for existence of PACN entrapment points is electrodiagnostic studies instead of cadaver dissection. Since there was no manual therapy result (nerve mobilization or traction) in this study, mentioning manual therapy techniques seemed totally irrelevant to cadaver results.

Results:

1.You mentioned three variations:

  1. ran close and parallel to the LIMS + PACN followed the posterior compartment (12/19)
  2. ran close and parallel to the LIMS + PACN crossed the LIMS with the RN. (4/19)

iii. clearly separated from RN before it crossed the LIMS (3/19)

A schematic picture showing these three variations with their respective percentage would be great.

2. There should be epicondylar branches and branch to the anconeus from P-PACN. Did you observe these branches in your cadaver, which were important for treatment rationale of tennis elbow? 

Limitations: Please move this paragraph to the end of the Discussion. Please explain “intrinsic technical constraints of US exploration.”

Conclusion: Please use one or two sentences to conclude your major findings in this study. Long paragraphs would distract readers’ attention.

References:

I didn't see the reference 36 in your main text. (Brian W, Starr MD, Dennis S et al (2020) Anatomy of the Posterior Antebrachial Cutaneous Nerve, Revisited. J Hand Surg 478 Am 45(4):360.e1-360.e4)

Figures

Figure 1: Please add “Arrow E” and “Arrow F” to show “E: The distance anteriorly/perpendicular where the A-PACN passes the LE F: The distance anteriorly/perpendicular where the P-PACN passes the LE” in Table 1.

Figure 2: The area you marked as “SF” looked like subcutaneous fatty layer. Please check.

Figure 4A: Please mark the RN and the separation point of PACN from RN.

Figure 8: This figure should be moved to the Results section.

Author Response

In this manuscript, the authors performed ultrasonography (US) and cadaveric studies, investigating the anatomical relationships of posterior antebrachial cutaneous nerve (PACN) with surrounding structures to identify possible critical points of compression. They found that PACN course maintains a close relationship with those of the radial nerve (RN), the lateral intermuscular septum (LIMS) and other connective tissues such as the deep and superficial fascia, and that the measurements obtained by US are in good agreement with further anatomical dissection measurement. I have several concerns for this manuscript.

 Thank you very much for your time and work.

Title:

“Can the course of the posterior antebrachial cutaneous nerve explain some elbow pains?” Your results did not answer this question.

We have rewritten the tittle. Thank you very much for your suggestion we hope that now it is clearer.

 Abstract:

Authors concluded the tight relationships could explain the existence of PACN entrapment points and the possibility of successful elbow pain treatment by applying manual therapy techniques. However, golden standard for existence of PACN entrapment points is electrodiagnostic studies instead of cadaver dissection. Since there was no manual therapy result (nerve mobilization or traction) in this study, mentioning manual therapy techniques seemed totally irrelevant to cadaver results.

Thank you very much for your observation. Our aim was to advice that the relation of the PACN with other connective tissues have to be present for the pathology and treatment of this nerve. We have changed the abstract and added this idea.

Results:

1.You mentioned three variations:

ran close and parallel to the LIMS + PACN followed the posterior compartment (12/19)

ran close and parallel to the LIMS + PACN crossed the LIMS with the RN. (4/19)

iii. clearly separated from RN before it crossed the LIMS (3/19)

A schematic picture showing these three variations with their respective percentage would be great.

Thank you very much for this suggestion we have added a schematic picture (Figure 5) to show this.

  1. There should be epicondylar branches and branch to the anconeus from P-PACN. Did you observe these branches in your cadaver, which were important for treatment rationale of tennis elbow?

Thank you very much for this observation.

We have found a great variability in the course of the PACN as we have described in the article. Sometimes this variability is difficult to explain, but in all our dissections we observed that the course of the anterior and posterior division of the PACN was always anterior to the lateral epicondyle as MacAvoy describes [17], and opposite to the Maida et al descriptions [18] that describe some examples of course of it he PACN posterior to the lateral epicondyle. Those, in these cases the skin of the anconeus could be innervate with this branches [18]

For this skin innervation in the lateral epicondyle that we found in our dissections we dare to say that some pain in this área can be by compression of this nerve.

Limitations: Please move this paragraph to the end of the Discussion. Please explain “intrinsic technical constraints of US exploration.”

Thank you very much for your observation, we have rewritten the paragraph. 

Conclusion: Please use one or two sentences to conclude your major findings in this study. Long paragraphs would distract readers’ attention.

You are right,  we are sorry, we have shortened this section.

References:

I didn't see the reference 36 in your main text. (Brian W, Starr MD, Dennis S et al (2020) Anatomy of the Posterior Antebrachial Cutaneous Nerve, Revisited. J Hand Surg 478 Am 45(4):360.e1-360.e4)

The reference has been removed.  

Figures

Figure 1: Please add “Arrow E” and “Arrow F” to show “E: The distance anteriorly/perpendicular where the A-PACN passes the LE F: The distance anteriorly/perpendicular where the P-PACN passes the LE” in Table 1.

We are sorry and you are right.  The distance anteriorly/perpendicular where the A-PACN passes the LE and F: The distance anteriorly/perpendicular where the P-PACN passes the LE” only is in the Table 1. We did not want to add more arrows to simplify the draw

Figure 2: The area you marked as “SF” looked like subcutaneous fatty layer. Please check.

Thank you very much; yes, we have signed SF as superficial fascia that it correspond to the subcutaneous fatty layer. We have added this in the text of the picture.

Figure 4A: Please mark the RN and the separation point of PACN from RN.

We are very sorry but in this picture we can not marked the RN because it is deep to the LHTB aponeurosis, that protects this nerve. We have rectified the text of the picture:

Posterior antebrachial cutaneous nerve (PACN) separated from the radial nerve that in the picture is deep to the LHTB longitudinal aponeurosis, just before it crosses the LIMS (16%) PACN ran alone inside the muscular fibres of the triceps brachii muscle in a canal formed by the perimysium and aponeurotic bands of this muscle. Lateral intermuscular septum (LIMS), lateral head triceps brachii muscle (LHTB), deep fascia (DF).

Figure 8: This figure should be moved to the Results section.

Thank you very much. It has been moved.

Reviewer 3 Report

With this study, the authors aimed to examine, either with Ultrasound and Dissection, the anatomical relationships of PACN with surrounding structures to identify eventual compression points. It is an interesting study. Some concerns are listed below.

Title: although I understand it, I am not convinced the title expresses the work performed. 

Abstract

The abstract should be coincident with the text. Please revise it accordingly.

Introduction

Make sure that the study aims are clear and can be answered by the study performed.

Methods

Please decide if the comparison between US and dissection is one of the study aims and in the case it is, present these results in the results section. The same occurs with the Bland-Altmann graphic.

Table 1 should be corrected.

Discussion

Include and discuss your study limitations in this section.

Limitations presented should be deeply described, and avoid statements that cannot result from your study.  

Conclusions

Conclusions should resume the main findings according to the study objectives. Try to clarify this section objectively.

Please correct some extra spaces and missing letters like in Line 150 “landmar:“ and Line 346 “reginal”

Author Response

Comments and Suggestions for Authors

With this study, the authors aimed to examine, either with Ultrasound and Dissection, the anatomical relationships of PACN with surrounding structures to identify eventual compression points. It is an interesting study. Some concerns are listed below.

Title: although I understand it, I am not convinced the title expresses the work performed.

Thank you very much for your observation. We have changed the title to reflect the work.

Abstract

The abstract should be coincident with the text. Please revise it accordingly.

Thank you very much. We have donned.

Introduction

Make sure that the study aims are clear and can be answered by the study performed.

Thank you very much for your suggestion, we have adapted in the aim our investigation.

Methods

Please decide if the comparison between US and dissection is one of the study aims and in the case it is, present these results in the results section. The same occurs with the Bland-Altmann graphic.

We are sorry but we believe that in the methods we have included that we injected where the radial nerve pierce the LIMS and where the PACN pierced the fascia (105-109) and we confirm it in the results (164, 177-180).

Table 1 should be corrected.

We have correct it. Thank you very much

Discussion

Include and discuss your study limitations in this section.

Limitations presented should be deeply described, and avoid statements that cannot result from your study. 

Thank you very much for the observation. We have change the limitations to the end of the discussion. We really would like to study the clinical implication of the fascial compression in patients to show how this connective tissue has an important paper in the nerve compression.

Conclusions

Conclusions should resume the main findings according to the study objectives. Try to clarify this section objectively.

Thank you very much for the observation. We have made a resume of the conclusions and we think that now it clarify this section.

Please correct some extra spaces and missing letters like in Line 150 “landmar:“ and Line 346 “reginal”

Thank you very much for the observation. We have made the corrections.

Reviewer 4 Report

Manuscript Title: Can the course of the posterior antebrachial cutaneous nerve explain some elbow pains? A cadaveric and sonographic study.

Overview: The aim of this study was to reveal the anatomical relatioships of PACN with surrounding structures along its pathway to identify posible critical points of compression

General comments: This is an interesting manuscript that addresses an important area of unmet medical need. Please see my specific comments below for more details.

Specific comments:

  1. Introduction: the Bibliographic references… not follow an order… start at number 24, please. This reviewer suspects that this is an error that needs to be corrected.
  1. The keywords are absolutely fine.
  1. Lines 120-121/199-200/220-221/231-232/260-261: images too large. This reviewer suspects that this is an error that needs to be corrected.

Author Response

Response of Authors to Reviewer #4

Overview: The aim of this study was to reveal the anatomical relatioships of PACN with surrounding structures along its pathway to identify posible critical points of compression

General comments: This is an interesting manuscript that addresses an important area of unmet medical need. Please see my specific comments below for more details.

Thank you for taking the time to review our submission and for providing some comments and suggestions that have helped us to improve the manuscript. We have amended the manuscript according to your suggestions and provided our comments in the section bellow. The amended sections of the manuscript are marked in red. Thank you in advance for your further perusal of our revision.

Specific comments:

  1. Introduction: the Bibliographic references… not follow an order… start at number 24, please. This reviewer suspects that this is an error that needs to be corrected.

Thank you, this has been corrected

  1. The keywords are absolutely fine.

Thank you, this has been corrected

  1. Lines 120-121/199-200/220-221/231-232/260-261: images too large. This reviewer suspects that this is an error that needs to be corrected.

A large size is required to appreciate minor anatomical details. However, this will be improved when the professional staff at MDPI will apply the final layout of the article.

Round 2

Reviewer 2 Report

The manuscript improved much in this version. The new title is more precise, and the new figure 5 is clear for readers. I have some concerns.

  1. After revision, this article focused on anatomy, sonography and clinical implications. Please consider to delete line 44-48 to make the introduction simpler, because this study is not limited to sports injury.
  2. In line 51-54, please delete References 14 and 16 because they seemed not related to your article. Since your study focused on the sonographic implications in compression of upper limb cutaneous nerves, please consider to cite some updated references such as “American Journal of Physical Medicine & Rehabilitation. 2018 Jul;97(7):e68”, “American Journal of Physical Medicine & Rehabilitation. 2019 Sep;98(9):e106”, and “Pain Medicine, 2020 Sep 21(9):2001–2002”.
  3. In line 90, “manual therapy” may be revised as “interventional therapy” or likewise, because your study is not limited to manual therapy.
  4. Figure 5: It would be clearer for readers if you add percentage on the figure. For example, please put “5%” besides the “A” mark, “10.5%” besides the “B” mark, “63.1%” besides the “C” mark, and “15.7%” besides the “D” mark.
  5. In original manuscript, you mentioned “intrinsic technical constraints of US exploration” in the Limitations section, but did not mention it in the revised manuscript. Please explain the reason.

Author Response

The manuscript improved much in this version. The new title is more precise, and the new figure 5 is clear for readers. I have some concerns.

Thank you for your words. The article improved thanks to you and the rest of the reviewers.

After revision, this article focused on anatomy, sonography and clinical implications. Please consider to delete line 44-48 to make the introduction simpler, because this study is not limited to sports injury.

Thank you. We rewrite this paragraph, and it is more general, clear and direct.

In line 51-54, please delete References 14 and 16 because they seemed not related to your article. Since your study focused on the sonographic implications in compression of upper limb cutaneous nerves, please consider to cite some updated references such as “American Journal of Physical Medicine & Rehabilitation. 2018 Jul;97(7):e68”, “American Journal of Physical Medicine & Rehabilitation. 2019 Sep;98(9):e106”, and “Pain Medicine, 2020 Sep 21(9):2001–2002”.

Thank you very much for these valuable references. We have added two of them.

In line 90, “manual therapy” may be revised as “interventional therapy” or likewise, because your study is not limited to manual therapy.

It has changed.

Figure 5: It would be clearer for readers if you add percentage on the figure. For example, please put “5%” besides the “A” mark, “10.5%” besides the “B” mark, “63.1%” besides the “C” mark, and “15.7%” besides the “D” mark.

Now the figure is more precise and has more information. Thank you very much.

In original manuscript, you mentioned “intrinsic technical constraints of US exploration” in the Limitations section, but did not mention it in the revised manuscript. Please explain the reason.

This sentence was disappeared at the suggestion of one reviewer. It has been added again because we think that it is a limitation of US.